# Impact of Protein Binding Capacity and Daily Dosage of a Drug on Total Serum Bilirubin Levels in Susceptible Infants

**DOI:** 10.3390/children10060926

**Published:** 2023-05-24

**Authors:** Zon-Min Lee, Ling-Sai Chang, Kuang-Che Kuo, Meng-Chiao Lin, Hong-Ren Yu

**Affiliations:** 1Department of Pharmacy, Kaohsiung Chang Gung Memorial Hospital, Kaohsiung 83301, Taiwan; zonmin@adm.cgmh.org.tw; 2Department of Pharmacy, Tajen University, Pingtung 907, Taiwan; 3Kawasaki Disease Center and Department of Pediatrics, Kaohsiung Chang Gung Memorial Hospital, Kaohsiung 83301, Taiwan; 8702034@adm.cgmh.org.tw; 4Division of Pediatric Infection, Kaohsiung Chang Gung Memorial Hospital, Kaohsiung 83301, Taiwan; light@cgmh.org.tw; 5Department of Pharmacy, St Joseph’s hospital, Yunlin 632401, Taiwan; st745@mail.stjoho.org.tw; 6Department of Pediatrics, Chang Gung Memorial Hospital-Kaohsiung Medical Center, Graduate Institute of Clinical Medical Science, Chang Gung University College of Medicine, Kaohsiung 83301, Taiwan

**Keywords:** drug dosage, protein binding, infant, total bilirubin, free bilirubin

## Abstract

Hyperbilirubinemia is a common pathological condition in neonates. Free bilirubin can penetrate the blood–brain barrier (BBB), which can lead to bilirubin neurotoxicity. In the context of predicting the risk of bilirubin neurotoxicity, although the specificity and sensitivity of free bilirubin levels are higher than those of total serum bilirubin (TSB), free bilirubin is not widely monitored in clinical practice. The threshold TSB levels at which phototherapy must be administered have been established previously. However, TSB levels are not well correlated with neurodevelopmental outcomes. Currently, TSB levels are commonly used to guide phototherapy for neonatal hyperbilirubinemia. Some clinical drugs can displace bilirubin from its albumin-binding sites, and consequently upregulate plasma bilirubin. Daily dosages play a vital role in regulating bilirubin levels. A drug with both a high protein binding capacity and high daily dosage significantly increases bilirubin levels in infants. Premature or very low birth weight (VLBW) infants are vulnerable to the upregulation of bilirubin levels as they exhibit the lowest reserve albumin levels and consequently the highest bilirubin toxicity index. Because bilirubin is involved in maintaining the balance between pro-oxidant and antioxidant agents, the downregulation of bilirubin levels is not always desirable. This review provides insights into the impact of protein binding capacity and daily dosage of drugs on the bilirubin levels in susceptible infants.

## 1. Introduction

Hyperbilirubinemia [1], characterized by elevated levels of total serum bilirubin (TSB), presents as yellowish discoloration of the skin, sclera, and mucous membranes in neonates [2,3,4]. This situation and a similar mechanism of pathogenicity also exists in mouse models [5]. Approximately 80% of premature infants and 50% of term neonates experience hyperbilirubinemia within the first weeks after birth, and almost 10% of infants continue to manifest increased levels of bilirubin up to 1 month of age [6,7]. Hyperbilirubinemia is commonly defined as a bilirubin level >15 mg/dL (>75th percentile in the first day) in term neonates [8]. In newborns, hyperbilirubinemia presents as jaundice and is frequently a benign condition. However, hyperbilirubinemia is a major cause of hospitalization in the first week of life [9], and the measurement of bilirubin levels is critical for the management of neonatal jaundice [10]. Bilirubin production in term neonates is 2 to 3 times higher than that in adults, which is mainly due to the decreased lifespan of red blood cells (RBCs) in neonates [11]. Compared with term neonates, premature infants exhibit a shorter RBC lifespan and consequently higher bilirubin production rates [3,12].

The balance between the production and excretion of bilirubin, Increased RBC breakdown, and/or decreased bilirubin excretion in neonates determines the severity of hyperbilirubinemia, as well as the risk of developing neurological dysfunction [3,12] and progression to acute bilirubin encephalopathy and kernicterus, which have a marked risk of neonatal mortality [9,13,14]. The development of hyperbilirubinemia into bilirubin neurotoxicity is avoidable with early detection and treatment. Screening for bilirubin levels [15] and the timely identification of risk factors that predispose infants to severe hyperbilirubinemia [16] are important to initiate phototherapy on time, and thus reduce mortality and morbidity. Common conditions [12] reported to increase hemolysis and bilirubin production in infants include RBC enzyme deficiencies, glucose-6-phosphate dehydrogenase (G6PD) deficiency, ABO immunization, RH isoimmunization, hereditary spherocytosis, sickle cell disease, thalassemia, sepsis, and ecchymosis.

G6PD deficiency and blood-type incompatibility deficiency are correlated with the development of neonatal hyperbilirubinemia [17,18,19,20,21,22]. Hemolytic disease of the newborn, which is caused by maternal antibodies attacking fetal RBCs (alloimmunization), may cause severe hyperbilirubinemia [23]. Various drugs are commonly administered for the treatment of diseases in neonates. Although most drugs are considered harmless, rare cases of drug-induced hyperbilirubinemia have been reported [24,25,26,27,28]. Kernicterus or bilirubin neurotoxicity may develop after free bilirubin crosses the blood–brain barrier (BBB) [24,29,30]. The development of bilirubin neurotoxicity is dependent on the ratio of unconjugated bilirubin concentration to reserve albumin concentration. This ratio increases when part of the albumin is occupied by a drug with apparent displacement of bilirubin [28]. Thus, the appropriate measurement of the serum bilirubin level is recommended for susceptible neonates. This review aims to provide insights into the impact of the protein binding capacity and daily dosage of a drug on bilirubin levels in susceptible infants.

## 2. Bilirubin and Its Binding with Albumin

Currently, TSB is the biochemical standard used to evaluate and manage neonatal hyperbilirubinemia [2]. Bilirubin, which is predominantly derived from hemoglobin degradation, is mainly bound to albumin in the blood, inhibiting its ability to cross the BBB [31,32]. Among the total bilirubin, not more than 0.01% constitutes circulating as free bilirubin (an unbound form) [33], and an even ratio of cord blood bilirubin/cord blood albumin at birth has been reported to be a good indicator for the prediction of neonatal hyperbilirubinaemia [34]. Bilirubin becomes water-soluble when it is bound to glucuronic acid during conjugation or albumin for transport in the serum, as well as when it is isomerized upon exposure to light [35].

Albumin functions as a vehicle for the transfer of bilirubin to the liver, where bilirubin dissociates from albumin and enters the hepatic cell for conjugation [31]. Hypoalbuminemia (a serum albumin <3.0 g/dL in term neonates an  <2.5 g/dL in preterm newborns) is a risk factor for bilirubin neurotoxicity, as low serum albumin levels may enhance the risk of bilirubin encephalopathy by lowering the bilirubin–albumin binding capacity [36,37]. Owing to the high affinity of albumin for bilirubin and the large quantity of albumin, the serum free bilirubin level is regulated by the albumin–bilirubin binding capacity in term newborns and healthy late preterm infants [31]. However, altered albumin–bilirubin binding capacity in premature infants dysregulates TSB [31]. Thus, conditions that affect the BBB function and bilirubin metabolism or increase brain blood flow need to be prevented [35]. If a drug competes with bilirubin to bind to albumin, it increases the concentrations of unbound bilirubin in the blood and promotes the transfer of an increased amount of bilirubin into the tissues, including the brain [35].

## 3. Phototherapy and Free Versus Total Bilirubin

Despite the fact that metalloporphyrins [38], Sn-mesoporphyrin [11], ursodeoxycholic acid [39], and phenobarbital [40,41] have been proposed as alternative approaches for preventing or treating neonatal hyperbilirubinemia, and melatonin promotes neuroprotective brain-derived neurotrophic factor expression in neonatal hemolytic hyperbilirubinemia [42], phototherapy is the current model. Phototherapy was first introduced in 1958 for the treatment of neonatal hyperbilirubinemia [43]. In a hospital setting, the blue part of the spectrum near the wavelength peak of 460 nm, is a safe and effective treatment without major side effects [44,45]. The indicator for phototherapy is a quickly increasing or high TSB level [6,46], which prompts non-polar unconjugated bilirubin in the neonatal skin converted to water-soluble isomers upon exposure to the blue light [38,45]. Phototherapy works by converting unconjugated bilirubin to the conjugated form, preventing brain damage caused by high levels of unconjugated bilirubin [47]. Along with the intrinsic characteristics of neonates, factors that can determine the efficacy of phototherapy include exposure duration, wavelength, distance from the skin, body surface area exposed to the light, and the type of phototherapy device [48]. 

Free (albumin-unbound) bilirubin is fat soluble, and the ability of premature infants to take up, conjugate, and excrete bilirubin is defective, resulting in a portion of the free bilirubin penetrating the BBB [45]. When large enough amounts of free bilirubin cross the BBB, they can form deposits in parts of the brain, causing severe neurotoxicity, which is closely correlated with bilirubin neurotoxicity [26,45,49]. Hence, the specificity and sensitivity of free bilirubin to determine the risk of bilirubin toxicity are higher than those of TSB levels [26,50,51]. Free bilirubin and TSB respond individually, and these two indices correspond well before phototherapy. Unlike free bilirubin, the levels of TSB tend to decrease after phototherapy [52]. Additionally, the significant displacement effect of an antibiotic on bilirubin–albumin binding may only be detected by examining the free bilirubin but not the TSB level [53]. However, free bilirubin is not widely monitored in clinical practice.

One study reported that extremely low birth weight (ELBW) neonates had developed kernicterus despite moderate management of unconjugated hyperbilirubinemia [54]. Although the threshold TSB levels at which phototherapy is administered have been established based on previous research, they are inconsistently correlated with the neurodevelopmental outcomes [55]. TSB levels are poorly correlated with bilirubin neurotoxicity and are not a sensitive or specific predictor of neurological outcomes [49]. Currently, the TSB level is commonly used to guide phototherapy for neonatal hyperbilirubinemia, and the use of phototherapy complies with protocols for the treatment of neonatal hyperbilirubinemia [56]. In hospital settings, TSB is monitored daily or less frequently to examine the risk for neurological dysfunction, initiate or stop phototherapy management, or initiate exchange transfusions in limited unresponsive cases.

Free bilirubin is a better indicator of neurotoxicity than TSB, as only free bilirubin can cross the BBB [51,57]. Currently the bilirubin/albumin (B/A) ratio is also considered as an alternative to identify the rare infant at risk for neurotoxicity at low TSB owing to a low serum albumin level [57]. The B/A ratio significantly correlates with free bilirubin levels in premature infants with a gestational age below week 35, and this ratio can be used as an index of free bilirubin levels in this group of infants, according to Abe’s study [58]. In one animal study [59], human serum albumin infusion was found to increase the plasma bilirubin binding capacity and the levels of circulating bilirubin from tissues to plasma, ultimately preventing severe neurological impairment in genetically modified mice. In another study, adjunct human serum albumin treatment was demonstrated to result in decreased brain bilirubin levels in phototherapy-treated Gunn rats by lowering free bilirubin in the plasma [60]. Generally, the detection of TSB, free bilirubin, albumin levels, B/A ratio, and albumin–bilirubin binding have been proposed for neonates with hyperbilirubinemia, with individual or combined uncertain predictive utility [19,56,58,61,62,63].

## 4. Impact of Protein Binding Capacity and Daily Dosage of a Drug

The risk of developing neurologic dysfunction in infants is dependent on the concentration of unconjugated bilirubin, the affinity of serum albumin to bind bilirubin, and the total amount of albumin [64]. The bilirubin/albumin molar ratio is high in infants with hyperbilirubinemia, which increases their susceptibility to the administration of competing drugs [65]. Some clinical drugs [24,26,28,65,66,67,68] can markedly displace bilirubin from its albumin-binding sites, and consequently upregulate plasma bilirubin, which can cross the BBB and cause neurotoxicity in infants [25]. Factors such as genetic variations and some pathological conditions also contribute to bilirubin neurotoxicity [64,69]. These findings suggest that the use of a drug that can displace bilirubin from its albumin-binding site upregulates free bilirubin and increases the risk for infants to develop neurological dysfunction [64]. Additionally, “the presence of binding competitors has minimal effect on drug distribution or the drug effect, except when the drug is tightly bound” [65]. This indicates that only drugs with a high protein binding ability can cause a significant upregulation of bilirubin levels in infants.

Highly bound is defined as >95% protein binding [70], while drugs with <75% protein binding are generally not considered to cause clinically significant effects [71]. The protein binding rates of furosemide [72] and sulfisoxazole [68] are 91–99% and 85%, respectively. The addition of furosemide displaced bilirubin from albumin globulins and RBCs in vitro [73], and furosemide, mole for mole, displaced as much bilirubin as sulfisoxazole [73]. Another study reported that bumetanide with a high protein binding rate of 94–96% [74] binds tightly to the protein in the cord serum [75]. Esomeprazole [76] and propranolol [77], whose protein binding rates are 97% and 93%, respectively, have been reported to significantly upregulate the total bilirubin levels in rats [78,79]. Thus, drugs with a high protein binding capacity exhibit a high displacing ability and consequently increase the level of bilirubin in vitro [80].

Although protein binding is a fundamental mechanism underlying the effect of a drug on TSB levels in infants [65], it is not the only mechanism. The protein binding rates of indomethacin [81] and furosemide [72] are 99% and 91–99%, respectively, while the daily dosages are only 0.2 mg/kg [81] and 1–2 mg/kg [82], respectively. Indomethacin and furosemide are constantly administered in neonates for the treatment of patent ductus arteriosus (PDA) and edema, yet these two drugs have never been reported to significantly increase bilirubin levels. In addition, furosemide does not produce a significant increase in free bilirubin in most infants [83]. Bumetanide [74] and esomeprazole [76], which have protein binding rates of 94–96% and 97%, respectively, are commonly administered at low daily dosages of 0.02–0.2 mg/kg for the management of fluid overload in neonates [84], and 0.5–1 mg/kg [85] for erosive esophagitis in infants, respectively [86]. However, these two drugs have not been reported to increase the bilirubin levels in infants. Propranolol [77] and amphotericin B [87], which have protein binding rates of approximately 93% and >90%, respectively, are constantly used at low dosages of 0.5–2 mg/kg/day [88] and 1 mg/kg/day [89] in neonates to treat congestive heart failure and infections individually. These two drugs are also not reported to be associated with neonatal hyperbilirubinemia (Table 1). These examples indicate that a significant upregulation of bilirubin levels in infants only occurs when a drug with both a high protein binding ability and a high daily dosage is administered, as evidenced by Cashore [90].

The daily dosage of a drug also plays a vital role in regulating the bilirubin levels [90]. The daily dosages of sulfisoxazole [91] and novobiocin [92] are 75–150 mg/kg/day and 15–30 mg/kg/day, while the protein binding rates are 85% [68] and >90% [93], respectively, for the treatment of infections in infants. These two antibiotics, although not considered highly bound to plasma protein, significantly upregulate the bilirubin levels and should not be used in infants with hyperbilirubinemia (Table 1) [26,27,102]. Furthermore, sulfisoxazole [103,104] and novobiocin [105,106] have also been proven to significantly increase TSB levels in animal studies, among which sulfisoxazole was found to increase free bilirubin levels in human infants [107].

Depending on the protein binding capacity and daily dosage, some drugs are considered safe and do not significantly upregulate bilirubin levels in infants. However, the administration of several drugs should be avoided in infants with hyperbilirubinemia (Table 1). The effects of various drugs administered in infants on bilirubin levels have drawn mixed conclusions. Cotrimoxazole [25], ibuprofen [108], ceftriaxone [94], and vitamin K [95] have been administered in infants to treat neonatal diseases for decades, yet increased bilirubin levels [66,102] or risk of kernicterus [68,96] have been reported due to the accumulation of drug dosages or the susceptibility of infants (Table 1).

## 5. Susceptible Groups: Premature or Very Low Birth Weight (VLBW) Infants

Disorders associated with impaired binding of bilirubin to albumin may result in neurological injury in premature infants (<37 weeks’ gestational age) with decreased bilirubin levels, leading to acute bilirubin encephalopathy in these infants without marked hyperbilirubinemia [64,109]. VLBW neonates (with a BW of <1.5 kg) are at a high risk of kernicterus or brain injury after hyperbilirubinemia [110], and VLBW infants have been reported to develop kernicterus caused by hyperbilirubinemia, which occurs outside the typical high-risk period [48]. In particular, premature or VLBW infants are highly vulnerable and have the highest bilirubin toxicity index mainly due to the lowest reserve albumin levels [54], low bilirubin binding capacity [111], and a high bilirubin/albumin molar ratio [31].

In spite of a low incidence in term infants, the occurrence of bilirubin encephalopathy in preterm infants of gestational age <30 weeks is 1.8 per 1000 live births [112]. Premature infants are at an increased risk of bilirubin neurotoxicity at lower levels of TSB than term infants during the first week of life [113]. The higher risk for bilirubin neurotoxicity is a consequence of the increased susceptibility of premature neuronal cells to bilirubin injury when free bilirubin crosses the intact BBB [113]. In addition, albumin levels are generally low in the postnatal period, which rise 15% in premature infants during the first 3 weeks [114]; hence, caution must be exercised in drug use within 3 weeks after birth.

High bilirubin levels may cause oxidative damage, followed by significant cytotoxicity and neurotoxicity in infants [115]. However, bilirubin is also associated with maintaining the balance between pro-oxidant and antioxidant agents [116]. Slight elevations in bilirubin levels promote antioxidant effects, consequently exerting neuroprotective effects in neonates [45,50,115,116,117]. Furthermore, aggressive phototherapy can increase mortality in some extremely low BW infants [12]. Thus, lowering the levels of bilirubin is not always desirable.

## 6. Discussion

The protein binding rate and daily dosage of cotrimoxazole (sulfamethoxazole + trimethoprim) for infants is 70% [68] and 50–60 mg/kg/day [68] Cotrimoxazole (sulfamethoxazole), for the treatment of infections in infants, is contraindicated for infants younger than 2 months [68]. One study reported that alternative drugs should be utilized in jaundiced or preterm infants, due to the risk of bilirubin displacement and kernicterus [68], while another study indicated that it is unlikely to cause kernicterus when administered in neonates at oral doses for 7–10 days (Table 1) [25]. Cotrimoxazole exhibits a medium protein binding rate and a high daily dosage. It is worth noting that its medium protein binding rate may the reason for the inconsistent results with cotrimoxazole. Additionally, the medium protein binding rate of cotrimoxazole may not promote the accumulation of free bilirubin to a level that can cross the BBB within 10 days. Furthermore, cotrimoxazole use for more than 10 days may lead to drug accumulation and increase the risk of kernicterus in neonates with hyperbilirubinemia. These issues will need to be examined in further studies.

The use of ibuprofen has been reported to increase bilirubin toxicity in rat cortical neuronal cultures [80]. However, the evaluation of the effect of ibuprofen on bilirubin levels in human infants has revealed inconsistent results. Several studies [66,102] found that ibuprofen upregulates bilirubin levels in neonates, whereas another study [108] demonstrated that “ibuprofen may not be associated with the bilirubin displacement effect in relatively stable premature infants with mild to moderate unconjugated hyperbilirubinemia”. Furthermore, ibuprofen was found to increase the free bilirubin levels in sick premature infants [118], but not in infants with mild hyperbilirubinemia [119]. A high protein binding rate (95%) [97] and a “moderate daily dosage” (5–10 mg/kg/day) [98] of ibuprofen may be the reason for the inconsistent results in infants (Table 1). Previously, we reported that “as ibuprofen use is correlated with apparent increase in TSB in neonates with a lower BW, especially in those with a BW <1 kg, i.v. acetaminophen can be an appropriate alternative to ibuprofen for extremely low BW neonates for the treatment of PDA if they are experiencing severe unconjugated hyperbilirubinemia” [67]. This conclusion is consistent with that of previous studies [64,110], suggesting that premature or/and VLBW infants are vulnerable to the drug-induced upregulation of bilirubin levels.

Ceftriaxone increased free bilirubin in high-risk jaundiced infants [120], but not in infants with only mild hyperbilirubinemia [121]. Ceftriaxone, with a protein binding rate of 85–95% [101] and a daily dosage of 50 mg/kg/day, should be avoided or significantly minimized in neonates with hyperbilirubinemia [120,122,123], due to a significant competitive interaction of ceftriaxone with bilirubin–albumin binding in neonates. However, ceftriaxone at a dosage of 25–50 mg/kg/day was administered in neonates with meningitis [94], and this drug is also not considered to be associated with a bilirubin-displacing effect in neonates with a mild unconjugated hyperbilirubinemia (Table 1) [121,124]. In this context, 50 mg/kg/day is viewed as maximum dose of ceftriaxone and should be avoided in neonates with severe hyperbilirubinemia; hence, daily dosage plays a vital role in the elevation of bilirubin levels, in addition to infants’ conditions.

Vitamin K has a protein binding rate of 95% [100], dose of 1 to 5 mg i.v. daily (about 1–2 mg/kg/day), and is commonly administered for a mean of 3.7 days in neonates [95] for the treatment of hemorrhagic disease in newborns, and only occasionally leads to slight elevations in bilirubin levels. However, the administration of vitamin K at a dose of 10 mg (about 4–8 mg/kg/day) for 3 days in premature infants is often associated with unconjugated hyperbilirubinemia and even kernicterus (Table 1) [96,101]. Again, an increased daily dosage is usually the cause for increased competitive interaction with bilirubin–albumin binding in neonates.

The previously mentioned drugs (indomethacin, furosemide, bumetanide, esomeprazole, propranolol, and amphotericin B) (Table 1), which have a high capacity for protein binding, exhibit the ability to displace and upregulate the bilirubin levels in vitro. Additionally, “the presence of binding competitors has minimal effects on drug distribution or drug effect except when the drug is tightly bound” [65]. Thus, only drugs with a high protein binding ability can result in the significant displacement of bilirubin. However, these drugs, with high protein binding rates and low daily dosages, have not been reported to be associated with neonatal hyperbilirubinemia. These results are consistent with those of Cashore [90]. Drug concentration and daily dosage are the factors that significantly reduce the reserve albumin–bilirubin conjugates and further upregulate the bilirubin levels in infants. Hence, the magnitude of the protein binding rate and daily dosage of a drug that can initiate significant displacement of bilirubin, leading to clinically significant hyperbilirubinemia in infants and increased risk of bilirubin neurotoxicity, must be determined. Based on our current understanding [25,26,66,68,70,71,75,84,93,96,100,102,108,120], we have come up with a risk stratification metric for the upregulation of bilirubin levels in infants through the integration of the protein binding rate and daily dosage of a drug (Table 2).

Severe hyperbilirubinemia is practically proposed as an infant who develops visible jaundice within the first 24 h after birth, when TSB levels increase 5 mg/dL/day, when peak levels are higher than the expected normal range (i.e., >95th percentile), and when clinically visible jaundice persists for more than 14 days [16]. Drugs belonging to zone A showed very significantly increased bilirubin levels in infants, drugs belonging to zone B only showed significantly increased levels, while zone C drugs showed a high probability of increased levels, zone D drugs showed a lower probability of increased levels, and zone E and F drugs showed non-significantly increased levels in infants (Table 2). Thus, a new drug, whose effects on bilirubin levels are unknown and belonging to zone A or B, should be avoided or replaced whenever possible, and a new drug belonging to zone C or D should be monitored for its effects on the bilirubin levels in premature and/or VLBW infants with severe hyperbilirubinemia. Further studies are needed to determine the precise dosage or mole ranges for neonates.

If a new drug with both a high daily dosage and high protein binding ability is administered concurrently with any drug belonging to zone A or B (or drugs with mixed conclusions such as the latter four drugs in Table 1), the risk stratification of this new drug should be upgraded one level due to potential competitive interaction for protein binding of these two drugs. However, since real-world data are currently lacking, further studies are needed to determine these levels of influence.

Du et al. [12] reported that impaired hepatic conjugation and/or excessive bilirubin production exacerbate the severity of neonatal hyperbilirubinemia. Premature or VLBW infants are susceptible because they exhibit increased bilirubin production rates, due to the decreased RBC lifespan [12]. Additionally, hepatic conjugation of bilirubin is limited during early postnatal life due to the low expression of the bilirubin-conjugating enzyme [12] and the decreased capability of the liver to uptake the conjugated bilirubin, resulting in the development of hyperbilirubinemia [125]. Premature or VLBW infants have the lowest reserve albumin levels and binding affinity, and consequently, the highest bilirubin toxicity index [54]. Hence, caution must be exercised in drug use, especially within the first three weeks after birth [114].

The precise level at which unconjugated hyperbilirubinemia promotes oxidative stress has not been clarified [115]. A previous study reported that drugs functioning as inhibitors or substrates of the transporters and/or enzymes may also be involved in bilirubin excretion, affecting the levels of bilirubin in the plasma [69]. Additionally, the influence of variations in genes encoding for these enzymes on hyperbilirubinemia has been elucidated, and variations of the UGT1A1 Gly71Arg or UGT1A1 TATA promoter are associated with an increased risk of neonatal hyperbilirubinemia [126]. Diseases, such as hypothermia, hypoxia, acidosis, asphyxia, sepsis, and hypoalbuminemia, may also increase TSB levels [31,55]. The roles of factors, other than protein binding capacity and daily dosage of a drug, which are the main focus of this review, cannot be ruled out and must be evaluated by future studies. Although infrequent bilirubin neurotoxicity events are essential, we do not recommend the cessation of drugs for the treatment of diseases in neonates. This review provides a guide for clinicians to assess drug use and avoid administering inappropriate drugs in susceptible infants.

## 7. Conclusions

The effect of a new drug with both a high daily dosage and high protein binding ability on the TSB levels and its potential bilirubin neurotoxicity must be carefully evaluated before administration in premature and/or VLBW infants with severe hyperbilirubinemia. A drug, whose effects on bilirubin levels are unknown and belonging to zone A or B, should be avoided or replaced whenever possible, and a drug belonging to zone C or D should be monitored for its effects on bilirubin levels until free bilirubin or other effective indicators are widely employed for clinically diagnosing bilirubin neurotoxicity.

## Figures and Tables

**Table 1 children-10-00926-t001:** Impact of protein binding rate and daily dosage of a drug on total serum bilirubin levels in infants [68,72,74,76,77,81,82,84,85,87,88,89,91,92,93,94,95,96,97,98,99,100,101].

Drug	Protein Binding Rate	Common daily Dosage in Infants	Impact of a Drug on TSB	References
indomethacin	99%	0.2 mg/kg/day	no	[81]
furosemide	91% to 99%	1–2 mg/kg/day	no	[72,82]
bumetanide	94–96%	0.02 to 0.2 mg /kg/day	no	[74,84]
esomeprazole	97%	0.5–1 mg/kg/day	no	[76,85]
propranolol	93%	0.5 to 2 mg/kg/day	no	[77,88]
amphotericin B	>90%	1 mg/kg/day	no	[87,89]
sulfisoxazole	85%	75–150 mg/ kg/day	yes	[68,91]
novobiocin	>90%	15–30 mg/kg/day	yes	[92,93]
sulfamethoxazole(cotrimoxazole)	70 %	50–60 mg/kg/day	mixed conclusion	[68]
ibuprofen	95%	5–10 mg/kg/day	mixed conclusion	[97,98]
ceftriaxone	85% to 95%	25 to 50 mg/kg/day	mixed conclusion	[94,99]
Vitamin K	95%	1–2 mg/kg/day and 4–8 mg/kg/day	mixed conclusion	[95,96,100,101]

no: drugs not reported to be associated with increased bilirubin in infants; yes: drugs associated with significant increased bilirubin levels and should be avoided in infants with hyperbilirubinemia; mixed conclusion: drugs with mixed conclusions towards significantly increased bilirubin levels in infants; TSB: total serum bilirubin.

**Table 2 children-10-00926-t002:** Risk stratification of a drug on total serum bilirubin levels in susceptible infants.

Daily Dosage /Protein Binding	>50 mg /kg/day	15–50 mg /kg/day	5–15 mg /kg/day	<5 mg /kg/day
>95%	A	B	C	D
75–95%	B	C	D	E
<75%	C	D	E	F

Zone A: very significantly increased bilirubin levels in infants; Zone B: significantly increased bilirubin levels in infants; Zone C: probably increased bilirubin levels in infants; Zone D: possibly increased bilirubin levels in infants; Zone E: and F: non- significantly increased bilirubin levels in infants.

## Data Availability

Micromedex database.

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
