# Peer review of "Impact of Protein Binding Capacity and Daily Dosage of a Drug on Total Serum Bilirubin Levels in Susceptible Infants"

_children, 2023, doi:10.3390/children10060926_

Round 1

Reviewer 1 Report

Authors performed a narrative review (please specify), to describe the impact of protein binding capacity and daily dosage of a drug on the bilirubin levels in at risk neonates.

The review appears well performed and discussed. It could be interesting for neonatology to evaluate the risk of jaundice in relation to drug administered every day in neonates. 

I have only some comments to improve the presentation of the manuscript

1.         the aim of the manuscript should be presented in the abstract too.

2.         To my opinion table 2, 3 and 4 should be unified. I suggest to add a column in which you specify if the drug it has been correlated to bilirubin level changes, not or mixed conclusion are performed. In addition, reference should be added in the specific column only with the number corresponding at the number at the end for the manuscript and not in the table legend. All drug in this new table should be mentioned in the section “Impact of protein binding capacity and daily dosage of a drug” and discussed in discussion section. It could be nice mention the pathology that needs the administration of these drugs in discussion section, underling the neonates at risk of bilirubin alteration. 

3.         M: Micromedex database and U: UpToDate are not necessary as references. Please delete in the text and add the references reporting it at the end of the manuscript, as Journal guidelines requests. 

4.         Lines 114, please improve the sentence specifying the “etc”.

English should be improved, to increase the readability of the manuscript. 

Author Response

to reviewer one

Authors performed a narrative review (please specify), to describe the impact of protein binding capacity and daily dosage of a drug on the bilirubin levels in at risk neonates.

 The review appears well performed and discussed. It could be interesting for neonatology to evaluate the risk of jaundice in relation to drug administered every day in neonates. 

I have only some comments to improve the presentation of the manuscript

  1. the aim of the manuscript should be presented in the abstract too.

A: OK. We have added this expression to the abstract as directed. (in red)

  1. To my opinion table 2, 3 and 4 should be unified. I suggest to add a column in which you specify if the drug it has been correlated to bilirubin level changes, not or mixed conclusion are performed. In addition, reference should be added in the specific column only with the number corresponding at the number at the end for the manuscript and not in the table legend. All drug in this new table should be mentioned in the section “Impact of protein binding capacity and daily dosage of a drug” and discussed in discussion section. It could be nice mention the pathology that needs the administration of these drugs in discussion section, underling the neonates at risk of bilirubin alteration. 

A: .OK. We have integrated these three tables into one table (Table one) as expected with One additional column “impact of a drug on TSB” to differentiate different results.

All drugs in this new table are mentioned in the section. (please see a new paragraph, in red) As expected, we added each drug’s indication individually (those in red), and the last 4 drugs (those with mixed conclusions) at Discussion section (paragraph 1-4).

  1. M: Micromedex database and U: UpToDate are not necessary as references. Please delete in the text and add the references reporting it at the end of the manuscript, as Journal guidelines requests. 

A: OK. We have improved that.

  1. Lines 114, please improve the sentence specifying the “etc”.

A: OK. We have completed that sentence, at the 1st paragraph of “Phototherapy and free versus total bilirubin). (in red)

Comments on the Quality of English Language

English should be improved, to increase the readability of the manuscript.

A: OK. One professor helped us further check the English grammar, and this manuscript has been edited previously (see the editing certificate)  

Reviewer 2 Report

The authors reviewed the literature and tried to classify the risk of increasing free bilirubin concentrations in neonates by stratifying the drugs. The results are presented in Table 5. There are at least three issues with using Table 5 to estimate the risk of a new drug.

1) The rationale for the drug dose thresholds of 5, 15, and 50 mg/kg is unclear. Theoretically, the drug should be classified by moles, not mass (mg). This could be a potential hazard in estimating the risk of new drugs from this stratified classification.

2) In clinical practice, patients are often prescribed more than one drug. There is no discussion regarding how to utilize this stratified classification in conditions where multiple drugs are used.

3) As the authors described, premature infants require a cautious approach. There is no discussion regarding how to utilize this stratified classification in premature infants.

Author Response

Reviewers 2

The authors reviewed the literature and tried to classify the risk of increasing free bilirubin concentrations in neonates by stratifying the drugs. The results are presented in Table 5. There are at least three issues with using Table 5 to estimate the risk of a new drug. 

  • The rationale for the drug dose thresholds of 5, 15, and 50 mg/kg is unclear. Theoretically, the drug should be classified by moles, not mass (mg). This could be a potential hazard in estimating the risk of new drugs from this stratified classification.

A: In Table 5 (now Table 2): Protein binding rates of 95% and 75% are based on the 2nd paragraph of “Impact of protein binding capacity and daily dosage of a drug”, in blue.

With the first 6 drugs in Table one are highly protein bound but commonly used in neonates without increased bilirubin levels, indicating dosage of 0.2-2mg/kg/day is quite safe. In addition, ibuprofen (Table one) (5-10mg/kg/day) draws mixed conclusion, suggesting 5mg/kg/day may be the boundary. Also, vitamin K (Table one) 1-2mg/kg/day is safe, but 4-8mg/kg/day associated unconjugated hyperbilirubinemia and even kernicterus (see 4th paragraph of Discussion)

Novobiocin (Table one) with protein binding >90 and 15-30mg/kg/day should be avoided in infants with hyperbilirubinemia), indicating 15mg/kg/day may be another boundary

Sulfisoxazole (Table one) 75-150mg/kg/day should be avoided in infants with hyperbilirubinemia. Besides, ceftriaxone (Table one) with protein binding 85% to 95%, and 25-50mg/kg/day is administered for neonates with meningitis, but daily dosage of 50mg/kg/day, should be avoided or significantly minimized in neonates with hyperbilirubinemia in some literatures, (See 3rd paragraph of Discussion). In addition, cotrimoxazole 50-60mg/kg/day causes mixed conclusion. (Table one) 50mg/kg/day could be the other boundary.

It seems that moles are more reasonable than daily dosage for comparison of competition of protein binding. However, this theory is based on one by one, meaning that one molecule of drug to one protein, how about one drug to two protein or two drugs to one protein? Who knows?

 In real world we could not make any conclusion based on moles due to incomplete data base currently. (Please see the original tables below.) In addition, a clinical doctor cannot easily understand or figure out the meaning if a drug expressed in mole

Based on available data, only drugs expressed in mg/kg/day can give a rough meaning for risk stratification which can help clinical doctors choose appropriate drugs. Further studies are needed to work out the precise dosage or mole ranges for neonates and revise this table. (See 5th and 6th paragraphs of Discussion) in blue.

drugs

Protein binding rate

( % )

Commonly

Used Dose

(mg/kg/d)

MW

( g )

µmole

bilirubin

displacer

acyclovir

  no

acetaminophen

10-25

30-60

151

298

no

albumin

  no

Adrenaline

  no

Amikacin

4-11

10-30

586

34

no

amoxicillin

  no

Amphotericin b

>90

0.5-1

924

  0.8

no

amiodarone

  no

Atropine

  no

Bumetanide

  no

carnitine

  no

Ceftriaxone

83-96

80-100

555

162

mixed

cefazolin

  no

Cefotaxime

27-38

100-200

455

330

no

captopril

  no

Cloxacillin

  no

clindamycin

  no

clonidine

  no

oral Sulfamethoxazole

70

50-60

253

217

Mixed

Colistin

  no

Calcium gluconate

  no

digoxin

  no

diazoxide

  no

diazepam

  no

doxycycline

  no

dopamine

  no

Erythromycin

  no

Fentanyl

  no

Furosemide

  no

Gentamicin

0-30

2.5-7.5

no

glucose

  no

Heparin

  no

Hydrocortisone

  no

Indomethacin

99

0.1-0.3

358

0.6

no

Insulin

  no

levothyroxine

  no

Lidocaine

  no

Methadone

  no

Metoclopramide

  no

Morphine

  no

naloxone

  no

netilmicin

  no

nystatin

  no

Parenteral nutrition

  no

Penicillin G

  no

meperidine

  no

phenobarbital

  no

Polymyxin B

  no

prednisolone

  no

Prostaglandin E1

  no

Rifampin

  no

Sodium benzoate

Preservative

Sodium bicarbonate

  no

Thyroxine

  no

Theophylline

  no

tobramycin

  no

vancomycin

  no

vitamin K 

mixed

Table One: related information of drugs commonly used in neonates

No.

Drugs

Protein binding rate

( % )

Commonly

Used Dose

(mg/kg/d)

MW

( g )

µmole

bilirubin

displacer

1

Amikacin

4-11

10-30

586

34

no

2

Ceftazidime

5-17

60-150

547

192

no

3

Fluconazole

11-12

6-12

306

29

no

4

Acetaminophen

10-25

30-60

151

298

no

5

Ampicillin

20

50-200

349

358

  no

6

Gentamicin

0-30

2.5-7.5

no

7

Cefotaxime

27-38

100-200

455

330

no

8

IM/IV carbenicillin

50

225-400

378

825

no

9

Moxalactam

52

100-200

520

288

no

10

IV aztreonam

56

60-90

435

172

no

11

IM /IV nafcillin

90

20-60

436

 92

  no

12

Amphotericin B

>90

0.5-1

924

  0.8

no

12

Oral/IV indomethacin

99

0.1-0.3

358

0.6

no

14

oral Sulfamethoxazole

70

50-60

253

217

Mixed

15

Sulfisoxazole

85

100

267

375

Yes

16

Ceftriaxone

83-96

80-100

555

162

mixed

17

Ibuprofen

99

15-30

206

109

Mixed

18

Novobiocin

95

30-60

613

73

yes

References

Micromedex

Micromedex

MI

Medline

  • In clinical practice, patients are often prescribed more than one drug. There is no discussion regarding how to utilize this stratified classification in conditions where multiple drugs are used.

A: For current available data, drugs proved “should be avoided in infants with hyperbilirubinemia” or “with mixed conclusions towards significantly increased bilirubin levels in infants” are presented in Table one.

  It is impossible to administer a new drug with drugs which should be avoided in infants with hyperbilirubinemia like sulfisoxazole or novobiocin (Table one), as these two drugs should not be used in neonates

If a new drug with both high daily dosage and high protein binding ability is administered concurrently with any drug belonging to zone A or B (or drugs with mixed conclusions like the latter four drugs in Table one), the risk stratification of this new drug should be upgraded one level due to potential competitive interaction for protein binding of these two drugs. As “real-world” data are lacking, further investigation is required to find out levels of influence. (We add this section at 7th paragraph of Discussion, in blue)

  • As the authors described, premature infants require a cautious approach. There is no discussion regarding how to utilize this stratified classification in premature infants.

A: 1. yes.

Disorders associated with impaired binding of bilirubin to albumin may result in neurological injury in premature infants (<37 weeks’ gestational age), with decreased bilirubin levels leading to acute bilirubin encephalopathy in these infants without marked hyperbilirubinemia. [64] [85]VLBW neonates (with a BW of <1.5kg) are at a high risk of kernicterus or brain injury after hyperbilirubinemia, [86] and VLBW infants develop kernicterus caused by hyperbilirubinemia that occurs outside the typical high-risk period has been reported. premature or VLBW infants are highly vulnerable and have the highest bilirubin toxicity index mainly due to the lowest reserve albumin levels, [54] low bilirubin-binding capacity, [87] and a high bilirubin/albumin molar ratio.[Please see 1st paragraph of “Susceptible groups: premature or very low birth weight (VLBW) infants”] in blue

  1. While a new drug belonging to zone A or B should be avoided or replaced whenever possible, and a drug belonging to zone C or D should be followed bilirubin levels in premature and/or VLBW infants with severe hyperbilirubinemia. (See 6th paragraph of Discussion) in blue.
  2. full-term neonates (GA> 37 weeks) or neonates with BW>1.5kg, especially those without apparent hyperbilirubinemia are not considered vulnerable (susceptible) groups in this manuscript.

Round 2

Reviewer 1 Report

.

.

Author Response

To reviewer one

  English has been improved by a professional editor.

 (Please see new certificate of English editing)

Reviewer 2 Report

It is widely recognized that free bilirubin penetrates the blood-brain barrier (BBB) and leads to bilirubin neurotoxicity. As the authors note, inhibition of the binding of bilirubin to albumin elevates free bilirubin, leading to neurotoxicity. This does not necessarily result in an increase in total bilirubin.

Unconjugated bilirubin is glucuronidated by UGT1A1 in the liver to conjugated bilirubin, which is excreted into the bile. UGT1A1 is the only enzyme involved in this glucuronidation and is known to be the rate-limiting step. Some drugs inhibit or compete with this UGT1A1 activity, resulting in an increase in unconjugated bilirubin and consequently an increase in free bilirubin. The authors have not fully discussed this mechanism, however, attempt to explain it exclusively in terms of protein binding rates. Moreover, Table 1 lists total bilirubin levels and does not describe the effect on free bilirubin levels.

The nature of each drug should be discussed comprehensively.

Therefore, it is considered a potential risk to evaluate the risk of new drugs using the table proposed by the authors.

Author Response

To reviewer two

It is widely recognized that free bilirubin penetrates the blood-brain barrier (BBB) and leads to bilirubin neurotoxicity. As the authors note, inhibition of the binding of bilirubin to albumin elevates free bilirubin, leading to neurotoxicity. This does not necessarily result in an increase in total bilirubin.

A: Yes. The specificity and sensitivity of free bilirubin levels are higher than those of total serum bilirubin (TSB) level. However, free bilirubin is not widely monitored in clinical practice. (Please see abstract, in red)

Despite the fact that TSB levels are not well correlated with neurodevelopmental outcomes, TSB level is currently used to guide phototherapy for neonatal hyperbilirubinemia in the hospital. TSB level is monitored every day or every other day for those with hyperbilirubinemia and it provides useful (although not exact or perfect) information regarding hyperbilirubinemia or neurotoxicity. That’s why we have reviewed many articles to get information if any given drug elevated free bilirubin or TSB level, or caused neurotoxicity. 

Unconjugated bilirubin is glucuronidated by UGT1A1 in the liver to conjugated bilirubin, which is excreted into the bile. UGT1A1 is the only enzyme involved in this glucuronidation and is known to be the rate-limiting step. Some drugs inhibit or compete with this UGT1A1 activity, resulting in an increase in unconjugated bilirubin and consequently an increase in free bilirubin. The authors have not fully discussed this mechanism, however, attempt to explain it exclusively in terms of protein binding rates.

Table 1 lists total bilirubin levels and does not describe the effect on free bilirubin levels.

A1: Yes, that’s true. The risk of neonatal hyperbilirubinemia may be increased by the variation of UGT1A1 Gly71Arg in Asia and Africa, as well as the variation of UGT1A1 TATA promoter in Asia and Europe. (ref: Roles of UGT1A1 Gly71Arg and TATA promoter polymorphisms in neonatal hyperbilirubinemia: A meta-analysis. Gene. 2020 Apr 30;736:144409. doi: 10.1016/j.gene.2020.144409. Epub 2020 Jan 30.)

OK, we add this expression in the final paragraph of DISCUSSION, in red

In this manuscript, we focused on the situation while a new drug’s information is still unknown regarding increased bilirubin levels or not. (Please see 6th paragraph of DISCUSSION, in red) We revise and make it clearer “While a new drug, whose effects on bilirubin levels are unknown and belonging to zone A or B, should be avoided or replaced whenever possible.......

This manuscript provides an easy-to-use conclusion (clue) for clinical doctors, not for scientists, to choose appropriate drugs for the treatment of premature infants with severe hyperbilirubinemia.

A2: OK. We searched through Pubmed and Medline databases, with key words of “free bilirubin” and “drug name” listed on table one, and found many journals in vitro or animal data. We could only find 4 drugs (not 12 drugs) which clearly described increased free bilirubin or not in human infants. (Our table one and two are based on human infants data) Hence we add directions if this drug increase free bilirubin level or not in 3rd and 4th paragraph of Impact of protein binding capacity and daily dosage of a drug, and 2nd 3rd paragraphs of DISCUSSION.

(In total these 4 drugs: furosemide, sulfisoxazole, ibuprofen and ceftriaxone)

The reason why some drugs have been proved to be associated with increased bilirubin levels or neurotoxicity in human infants, yet drugs’ information as to “ increased free bilirubin or not “ is not easy to find, as follows

Slow infusion of sulfisoxazole caused a gradual and eventually pronounced decrease of total bilirubin concentrations in plasma but had no apparent effect on the concentration of free bilirubin at any time. These results are consistent with recently developed pharmacokinetic theory according to which the plasma clearance of total bilirubin should increase upon administration of a displacing agent while the plasma clearance of free bilirubin should remain unchanged. Bilirubin-induced encephalopathy caused by sulfisoxazole or other displacing agents may be due to very transient elevations of free bilirubin concentrations in plasma of infants with elevated plasma concentrations of total bilirubin and the consequent redistribution of the pigment to extravascular sites, including the brain.(ref: J Pharm Sci 1979 Jan;68(1):6-9. doi: 10.1002/jps.2600680106.)       

Moreover, The nature of each drug should be discussed comprehensively. Therefore, it is considered a potential risk to evaluate the risk of new drugs using the table proposed by the authors.

A: Yes, scientists have to be serious and take every drug individually doing research to prove if this new drug increases bilirubin level in infants or not.  

We did mention factors contributing to increased bilirubin levels. (Please see 1st paragraph of Impact of protein binding capacity and daily dosage of a drug, and final paragraph of DISCUSSION, in red)

However, this manuscript focuses on the situation that this new drug’s information is still unknown regarding increased bilirubin levels or not. (Usually it takes years or more than 10 years post a drug used in neonates) We published one article last year. Increased Total Serum Bilirubin Level Post-Ibuprofen Use Is Inversely Correlated with Neonates' Body Weight. Children (Basel), 2022. 9(8).

As both protein binding and daily dosage have been proved to be the major factors associated with increased bilirubin levels. Therefore, we come up with a risk stratification for the upregulation of bilirubin levels in infants, through the integration of protein binding rate and daily dosage of a drug. This table can provide a basic concept and clues, although not precise, for clinical doctors while treating susceptible group (premature or very low BW infants) with severe hyperbilirubinemia; reason 1: most doctors do not have any idea about relations between a drug and increased bilirubin, reason 2: It is easy to check a new drug’s protein binding rate (such as Micromedex) to use this table.

Round 3

Reviewer 2 Report

This revision is now a scientifically acceptable description.